



# First results of the XBAER aerosol optical depth algorithm with EnMAP data

Simon Laffoy[1], Marco Vountas[1], Linlu Mei[2, 3], and Hartmut Bösch[1]

[1]Institute of Environmental Physics, University of Bremen, Bremen, Germany.
[2]International Research Center of Big Data for Sustainable Development Goals, Beijing, China.
[3]Key Laboratory of Digital Earth Science, Aerospace Information Research Institute, Chinese Academy of Sciences, Beijing, China.

**Correspondence:** Simon Laffoy (slaffoy@iup.physik.uni-bremen.de)

**Abstract.** New high-resolution hyper and multispectral satellite instruments enable the retrieval of aerosol optical depth (AOD) at spatial resolutions of tens of meters. The eXtensible Bremen AErosol Retrieval (XBAER) AOD retrieval algorithm has previously been developed for use with Ocean and Land Colour Instrument (OLCI) and MEdium Resolution Imaging Spectrometer (MERIS) radiance data. With the intention of later modifying XBAER to use the full 30 m spatial resolution data from the

Hyper-Spectral Imager (HSI) on board the Environmental Mapping and Analysis Program (EnMAP) satellite, the present study investigates how HSI data compare to OLCI data. For the bands of interest, top of atmosphere reflectances generally compare well ($R > 0.9$), the intercept of the best fit line is less than 0.05 from the origin, and the slope is less than 0.1 from 1. However exceptions exist and these are explained as the result of differences in the spectral response functions of the instruments in the region of the spectrum around the O2 A-Band absorption feature, or as a result of differences in the viewing

geometry of the satellites which produces differing bidirectional reflectance distribution function (BRDF) effects. XBAER is then used to retrieve OLCI and HSI surface reflectance (SRF) and AOD. For SRF the comparison between OLCI and HSI yields $R = 0.953$, best fit intercept $= 0.003$ and best fit slope $= 1.082$. The respective comparison for AOD yields $R = 0.809$, best fit intercept $= 0.153$ and best fit slope $= 0.785$. These comparisons are then separated by surface type and insights are gained into the performance of the algorithm. Finally, the unmodified XBAER algorithm is run using the full spatial resolution

HSI data. Plumes from biomass-burning are identified in a single scene, and a comparison with AErosol RObotic NETwork (AERONET) AOD is performed for multiple scenes, achieving R = 0.631. Future modifications to XBAER that would allow it to produce more accurate retrievals at HSI's spatial resolution are discussed.

## 1 Introduction

Aerosol optical depth AOD is a measure of how much atmospheric aerosols weaken the transmission of radiation through the

atmosphere, and thus acts as a proxy measure for the amount of aerosol in the atmosphere. Retrieving AOD thus helps us to understand how aerosols affect climate, weather and health.

The most accurate measurements for AOD come from sparse surface-based measurements, such as the various sites of the AErosol RObotic NETwork (AERONET) (Holben et al., 1998). Airborne instruments can provide extended spatial coverage,



for example AOD has been retrieved using the Airborne Visible/Infrared Imaging Spectrometer (AVIRIS) data (Isakov et al.,
1996), but such measurements can only be performed for infrequent campaigns.

Several decades of AOD measurements from satellites are now available that offer increased spatial coverage over both
surface-based and airborne measurements. The spatial resolution of a satellite instrument's measurements gives an upper limit
on the spatial resolution of an AOD retrieval. For example the Moderate Resolution Imaging Spectroradiometer (MODIS)
has a spatial resolution of 250 m or 1000 m depending on the band. The Dark Target (Levy et al., 2013; Gupta et al., 2018),
Deep Blue (Hsu et al., 2013), and Multi-Angle Implementation of Atmospheric Correction (MAIAC) (Lyapustin et al., 2018)
algorithms have all been applied to MODIS data and obtained spatial resolutions of 3 km, 10 km and 1 km, respectively. The
Advanced Very High Resolution Radiometer (AVHRR) has a spatial resolution of about 1.1 km at nadir and Deep Blue has
been applied to its 4 km spatial resolution Global Area Coverage dataset to obtain AOD at 8.8 km spatial resolution (Hsu
et al., 2017). The Advanced Along-Track Scanning Radiometer (AATSR) produces data with a spatial resolution of 1 km and
the combined AATSR Dual-View (ADV) and AATSR Single-View (ASV) algorithms can produce AOD at the full AATSR
resolution of 1 km (Kolmonen et al., 2016).

While these AOD datasets are invaluable, higher resolution AOD can improve our understanding of its spatial distribution
which can help for example to improve exposure studies of health effects of aerosols. New multi and hyperspectral satellite
instruments with higher spatial resolution allow for increased spatial-resolution of an AOD retrieval. A non-exhaustive list of
examples includes data from the 2019-launched hyperspectral PRecursore IperSpettrale della Missione Applicativa (PRISMA)
satellite being used to retrieve AOD of industrial plumes at sub 100 m spatial resolution (Calassou et al., 2024). The Multi-
spectral Instrument (MSI) of Sentinel-2 has being used to retrieve 60 m AOD for urban areas (Yang et al., 2021). Landsat 8
data has been used to retrieve AOD at 30 m spatial resolution for dark pixels over Beijing (Ou et al., 2017).

The eXtensible Bremen AErosol Retrieval (XBAER) algorithm (Mei et al., 2017a, b, 2018) retrieves AOD, surface re-
flectance (SRF), and cloud parameters such as cloud optical thickness and cloud effective radius from satellite measurements. It
was originally developed for use with data from the MEdium Resolution Imaging Spectrometer (MERIS) (Mei et al., 2017a, b),
before being adapted to data from the Ocean and Land Colour Instrument (OLCI) (Mei et al., 2018). Although both of these
instruments have a spatial resolution of up to 300 m, XBAER used the reduced resolution datasets with spatial resolution of 1.2
km to produce data products with a spatial resolution of 10 km. The Hyper-Spectral Imager (HSI) on board the Environmen-
tal Mapping and Analysis Program (EnMAP) satellite (Chabrillat et al., 2024; Storch et al., 2023) has a much higher spatial
resolution of 30 m. This paper investigates the possibility of adapting the XBAER algorithm for use with HSI data.

In Section 2 the XBAER algorithm and the MERIS, OLCI and HSI instruments are introduced. In Section 3, as a first
step, the radiometric calibration of HSI is investigated by comparing its top of atmosphere reflectances (RTOA) to OLCI's for
colocated scenes at OLCI's spatial resolution. In Section 4 XBAER is run using the same colocated HSI and OLCI scenes
from the previous section, and the resulting SRF and AOD retrievals are analysed. First results of using the full 30 m HSI
spatial resolution with XBAER to retrieve AOD are presented in Section 5, including examination of biomass burning in a
single scene, and a comparison with AERONET data for multiple scenes. In Section 6 the challenges of further developing the
XBAER algorithm for use with HSI data are discussed.





## 2   Methods and Data

The XBAER algorithm performs retrievals of AOD as well as surface and cloud parameters (Mei et al., 2017a, 2019). It uses RTOAs derived from radiances measured in the visible and near-infrared (VNIR) range. It was originally developed using reduced spatial resolution (1.2 km) MERIS data achieving outputs at a spatial resolution of 10 km.

The XBAER retrieval begins by applying a cloud mask (Mei et al., 2017b) to minimize cloud contamination in AOD retrieval. Clouds are identified by comparing measures of scene-brightness, RTOA homogeneity and cloud height information
to calibrated threshold values. The threshold values are determined by a combination of radiative transfer modeling and an analysis of different cloud, aerosol and surface scenarios. Cloud adjacent pixels are also screened.

Once clouds are screened, SRF and AOD can be determined simultaneously in an iterative procedure. The surface contribution is determined by a linear function in one variable, where the variable is the soil-adjusted vegetation index (SAVI) and the parameters are determined by a generated dataset of spectral coefficients with spatial resolution of $0.1° \times 0.1°$ and monthly
temporal resolution. The surface contribution can then be separated from the RTOA using the Chandrasekhar equation relating RTOA and surface bidirectional properties (Mei et al., 2017a; Kaufman et al., 1997). Thus AOD may be retrieved using lookup tables of aerosol parameterisation generated by radiative transfer modeling . Subsequent versions of XBAER allow retrievals of cloud optical thickness, cloud effective radius and aerosol above cloud for aerosol-contaminated cloudy scenes.

MERIS was a pushbroom spectrometer with 15 spectral bands from 412 nm to 900 nm. It was on board the ENVISAT
satellite which flew in a sun-synchronous orbit. Contact was lost with it in 2012. The OLCI instrument (also a pushbroom spectrometer) on Sentinel-3 (also in a sun-synchronous orbit) is a successor to MERIS and XBAER was subsequently adapted for use with OLCI (Mei et al., 2018). OLCI has 21 spectral bands from 400 nm to 1020 nm, 14 of which have the same centers and full width at half maximums (FWHM) as MERIS bands. The recreation of the 15th MERIS band (MERIS Band 11) at the oxygen A-Band (O2 A-Band) absorption feature differs in center and FWHM by 0.625 nm and 1.25 nm respectively. The OLCI
swath width is 1270 km and crosses the equator at approximately 10 am local time. There are two OLCI instruments, A and B, onboard Sentinel-3A and Sentinel-3B respectively. Between the two satellites there is almost daily global coverage (some regions in the tropics are missed). Although small differences in spectral response functions exist between the two instruments, this paper treats them as interchangeable. OLCI spatial resolution is 300 m, but there is also a reduced resolution product of 1.2 km. This paper and XBAER use the reduced resolution Level-1B data product.

HSI on board the EnMAP satellite is a pushbroom hyperspectral imager and measures radiance data at 30 m, ten times the full spatial resolution of OLCI. A HSI scene actually consists of two almost overlapping scenes; the VNIR scene has 91 bands from 418 nm to 993 nm and the shortwave infrared (SWIR) scene has 133 bands from 901 nm to 2445 nm. As XBAER only uses wavelengths within the VNIR, this study uses HSI's Level 1B VNIR data. Finally, note that EnMAP is also in a sun-synchronous orbit and crosses the equator at approximately 11 am local time, an hour later than Sentinel-3 and OLCI.



| OLCI bands | | | Nearest HSI bands | | | Usage in XBAER | | |
|---|---|---|---|---|---|---|---|---|
| Band no. | Central wavelength | FWHM | Band no. | Central wavelength | FWHM | Cloud mask | Surface reflectance | AOD |
| 2 | 412.5 | 10 | 1 | 418.4 | 7 | x | x | x |
| 3 | 442.5 | 10 | 6 | 444.7 | 6.1 | | x | x |
| 4 | 490 | 10 | 16 | 491.8 | 5.8 | | x | x |
| 5 | 510 | 10 | 20 | 510.8 | 5.9 | | x | x |
| 6 | 560 | 10 | 30 | 561.1 | 6.5 | | x | x |
| 7 | 620 | 10 | 41 | 622.9 | 7.2 | | x | x |
| 8 | 665 | 10 | 48 | 666.6 | 7.7 | | x | x |
| 10 | 681.25 | 7.5 | 50 | 679.7 | 7.8 | | x | x |
| 11 | 708.75 | 10 | 54 | 706.6 | 8.1 | | x | x |
| 12 | 753.75 | 7.5 | 61 | 756.4 | 8.7 | x | | |
| 13 | 761.25 | 2.5 | 62 | 763.7 | 8.8 | x | | |
| 18 | 885 | 10 | 78 | 887.7 | 10 | | x | |

**Table 1. Band information.** OLCI bands used in XBAER, their central wavelength and full width at half maximum (FWHM), equivalent information for the nearest HSI band, and what that band is used for in XBAER.

## 3 Comparison of OLCI and HSI RTOA

To verify the calibration of HSI data for its use in XBAER it must be determined that HSI radiance data are consistent with data of previous satellite instruments used with XBAER, in this case OLCI. This must be done for every band used as input for XBAER, see Table 1 for a list of those bands.

Overlapping OLCI and HSI scenes for a selection of surface types (36 vegetation scenes, 50 desert and 45 urban) are identified. Note that whereas an OLCI scene is an entire descending node of an orbit, with the entire swath, a HSI scene is $1000 \times 1024$ pixels of area approximately 900 km$^2$ (30 km $\times$ 30 km). Throughout this paper, references to an OLCI scene that is colocated with a HSI scene, means only the small portion of the OLCI scene that overlaps with the much smaller HSI scene. The locations of the scenes are shown in Fig. 1.

For each 1.2 km OLCI pixel fully within a HSI scene, as much as 1,600 30 m HSI pixels may overlap it either fully or partially. Due to differences in angle and orientation of the instruments, these HSI pixels will not form a perfect square, but they will approximate one with up to 40 pixels to a side. Thus, 160 pixels of the 1,600 would overlap the edge of an OLCI pixel, meaning approximately 90% of these HSI pixels will be fully within that OLCI pixel. A weighted mean RTOA of the 30 m HSI pixels intersecting that OLCI pixel is calculated, weighted by the proportion of corners of each HSI pixel inside the OLCI pixel. If all four corners of a HSI pixel are within the OLCI pixel, then that pixel is assumed to be entirely within the OLCI pixel, and is assigned the maximum weight of 1. If only 3, 2 or 1 corners of a HSI pixel are within the OLCI pixel, then the proportion of the HSI pixel within the OLCI pixel is assumed to be within the range of 50% to 100%, 0% to 100% or 0%





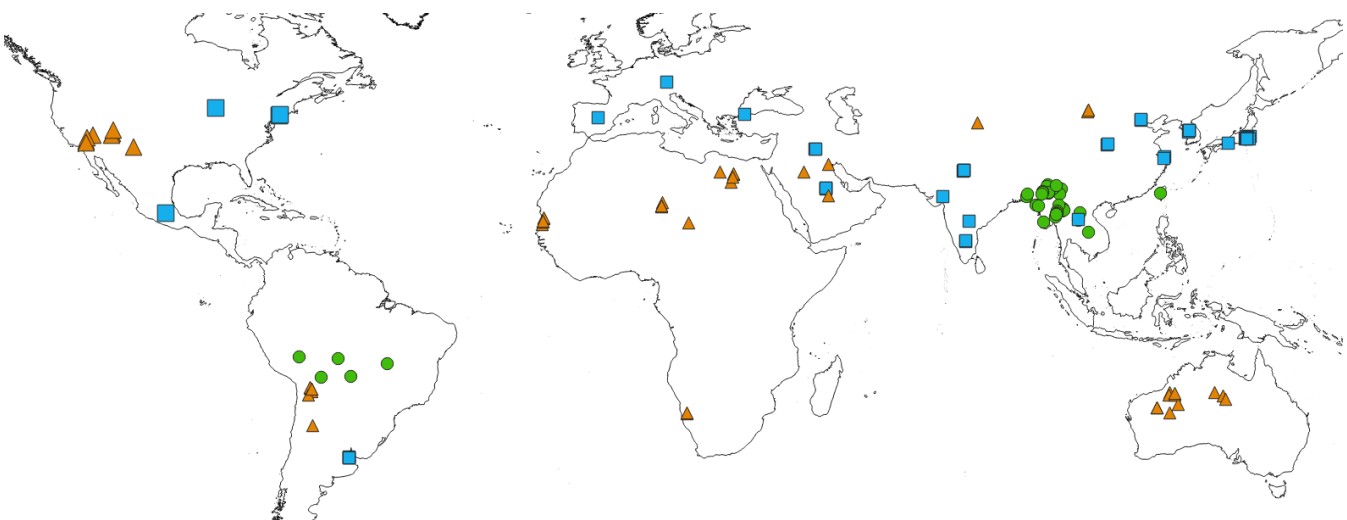

**Figure 1. Scene locations.** Locations of scenes chosen for RTOA comparison. Green circles are vegetation, orange triangles are desert, blue squares are urban.

to 50% respectively, and the midpoint of those ranges (0.75, 0.5 and 0.25) are assigned as the weight. In this way, a HSI RTOA is obtained for a 1.2 km pixel colocated with the OLCI pixel.

When processing a HSI and OLCI scene in the way just described, all OLCI pixels which are fully within the HSI scene

are used. This includes pixels for surface types that differ from the general scene surface type, for example settlements in predominantly vegetation scenes, rural hinterland in urban scenes and small or peripheral water bodies in all surface types.

HSI and OLCI measure radiances using different bands and different spectral response functions. For each OLCI band, a HSI band must be selected to compare to it. A statistical analysis of the performance of various band selection or composite band construction methods was performed. The five methods compared were 1) nearest HSI band to an OLCI band, 2) mean

of HSI bands overlapping an OLCI band, 3) weighted mean of HSI bands overlapping an OLCI band, weighted by amount of overlap, 4) and 5) linear regression of HSI bands overlapping an OLCI band, with the intercept term ineligible and eligible for regression respectively. Note that here a HSI band and an OLCI band overlap if their FWHMs intersect. It was determined that method 1) the nearest HSI band to an OLCI band performed as well, or nearly as well, as other more sophisticated methods. Going forward, this method is used. See Table 1 for details of which HSI bands are nearest to each OLCI band.

Scenes which are cloud-free at the time of both satellite overpasses were determined as follows. If one or both scenes are cloudy or partially cloudy, then a very poor RTOA comparison is produced for all wavelengths, or just some wavelengths if the cloud is thin. Visual inspection of a scene can then be performed to confirm the presence of clouds, and such scenes are then discarded. If visual inspection fails to identify clouds, then the scene is not discarded.

Presented in Fig. 2 is the OLCI vs HSI RTOA comparison for all 12 bands in Table 1 and all 50 desert scenes. Note that

the contribution to the RTOA from Rayleigh scattering (which overwhelmingly affects the lower wavelength bands) has been subtracted from the RTOAs (Fröhlich and Shaw, 1980). Going forward, when referring to the RTOA comparison for band i





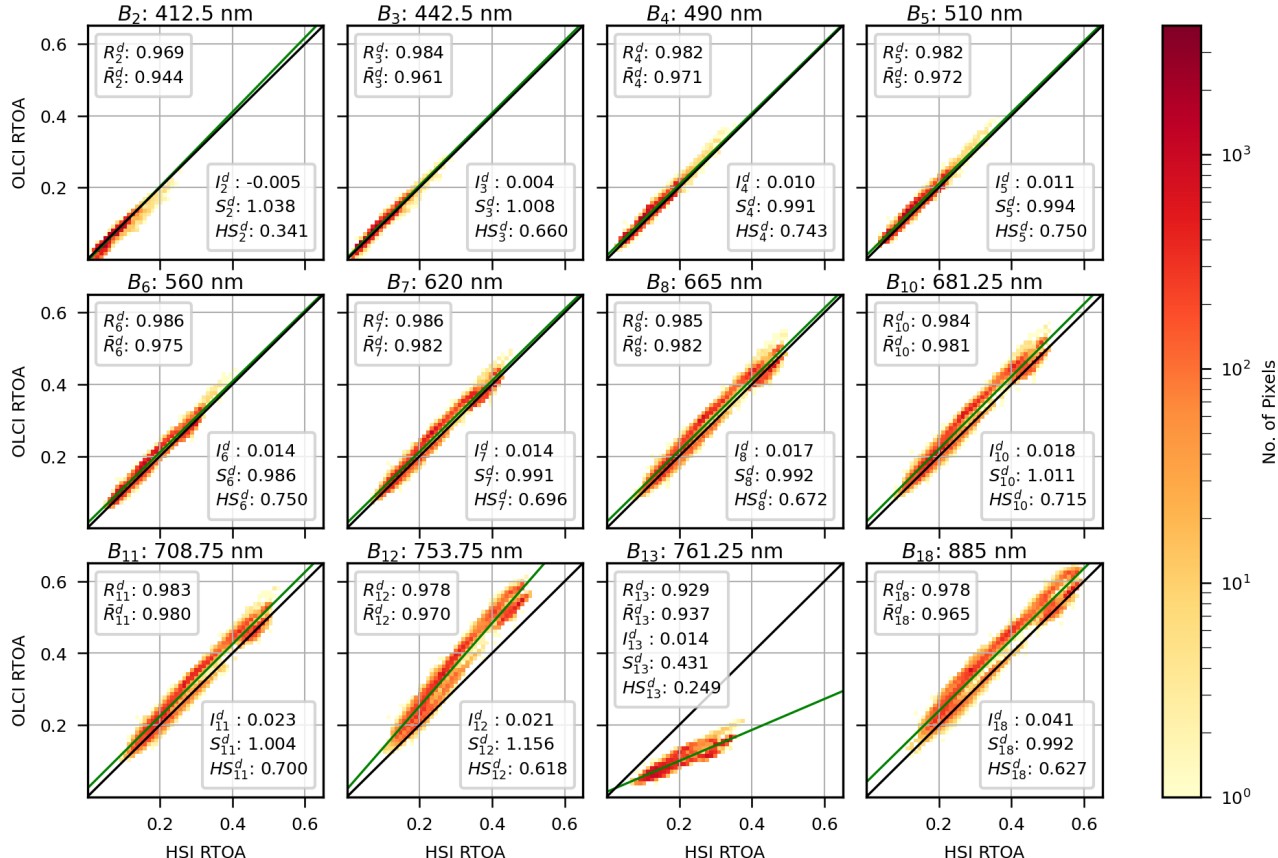

**Figure 2. Desert scene RTOA comparison.** RTOA comparison for 50 desert scenes. If $R_i^{d,j}$ represents correlation of RTOA between OLCI band i ($O_i$) and the nearest HSI band for all pixels in scene j, then $R_i^d$ is the correlation of all pixels in all scenes at band i, and $\bar{R}_i^d$ is the mean of $R_i^{d,j}$ for all j. $I_i^d$ and $S_i^d$ represent the intercept and slope of the green best fit line.

($B_i$), this means the comparison of RTOA for OLCI band i ($O_i$) with the RTOA for the nearest HSI band. $R_i^d$ is the Pearson correlation coefficient of RTOA for all pixels in all desert scenes for $B_i$. If $R_i^{d,j}$ represents the RTOA correlation for $B_i$ and desert scene j, then the mean of $R_i^{d,j}$ for all j is represented by $\bar{R}_i^d$. For each $B_i$, the best fit for all pixels in all scenes is depicted

as a green line, and the intercept and slope of this line are denoted by $I_i^d$ and $S_i^d$.

With the exception of $B_{12}$ and $B_{13}$, all other bands have an excellent comparisons with $R_i^d > 0.96$, $|I_i^d| < 0.05$ and $|S_i^d - 1| <$ 0.05. $B_{13}$ has the worst comparison with $S_{13}^d = 0.431$. As can be seen in Table 1, HSI band 62 ($H_{62}$) is the closest to $O_{13}$, and is significantly wider. Moreover, the narrow $O_{13}$ is centered near the deepest trough of the O2 A-Band, meaning that $H_{62}$ will measure higher radiances, see Fig. 3 (b). To a lesser extent the O2 A-Band also affects the RTOA comparison for $B_{12}$. $O_{12}$ does

not intersect the absorption feature, but $H_{61}$ does slightly, see Fig. 3 (a). Thus it is expected that $H_{61}$ measures lower radiances than $O_{12}$, and this is seen in $S_{12}^d = 1.156$ and the greater than zero $I_{12}^d = 0.021$. Despite the poor slopes of the comparison for $B_{12}$ and $B_{13}$, they have strong correlation and intercept close to 0; $R_{12}^d = 0.978$, $R_{13}^d = 0.929$, $I_{12}^d = 0.021$ and $I_{13}^d = 0.014$.





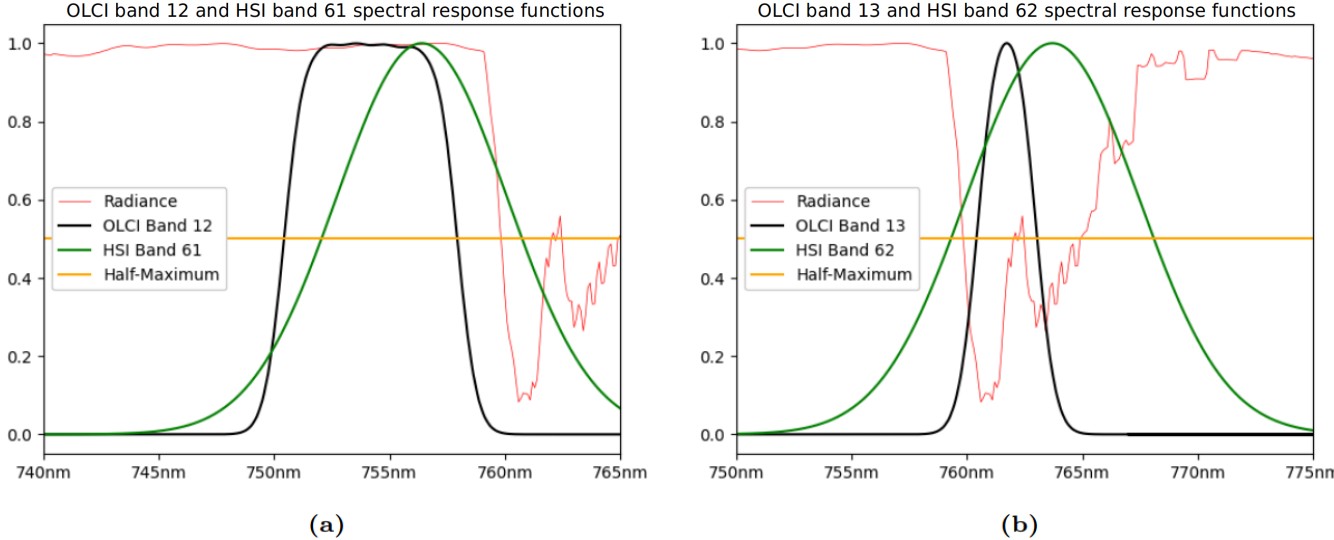

**Figure 3. O2 A-Band.** The spectral response functions of (a) $O_{12}$ and $H_{61}$ and (b) $O_{13}$ and $H_{62}$, overlain on a 1 nm rolling-average of radiances simulated using SCIATRAN (Rozanov et al., 2014) at 0.1 nm spectral resolution (red line). All curves are plotted relative to their maximum within the range. The orange line is intended as a visual aid to help identify the FWHM of the two spectral response functions.

$O_{12}$ and $O_{13}$ are only used in XBAER as part of the cloud mask. To be more specific, their ratio is compared to a threshold value and used as an indicator of cloud top height. As such, it is planned to modify the XBAER cloud mask threshold values for use with HSI data. Thus these poor comparisons for $B_{12}$ and $B_{13}$ are not considered to be a problem for adapting XBAER for HSI data.

The above analysis of 50 desert scenes is repeated for 45 urban scenes and 36 vegetation scenes, see Figs. 4 and 5. For urban scenes $R_i^u > 0.94$ for all bands, with the 3 uppermost bands ($B_{12}$, $B_{13}$, $B_{18}$) the only bands below 0.96. However the slope is not as good; only for $B_3$ is it within 0.05 of 1. For 6 bands it is more than 0.1 from 1 with bands $S_{12}$ (1.383), $S_{13}$ (0.508) and $S_{18}$ (1.250) performing worst. The explanation of the poor slope for $B_{12}$ and $B_{13}$ of desert scenes also applies to other scene types, but $S_{12}^u$ is much higher than $S_{12}^d$ (1.156) and some other effect may be occurring.

The vegetation scenes (Fig. 5) perform worse than urban scenes, with R above 0.95 for only two bands ($R_2^v = 0.956$, $R_3^v = 0.959$), and below 0.8 for all upper bands ($R_{11}^v = 0.774$, $R_{12}^v = 0.614$, $R_{13}^v = 0.780$, $R_{18}^v = 0.583$) However, despite this, mean scene correlation, $\bar{R}_i^v$ remains everywhere above 0.9. Thus pixels within a scene are usually very well correlated, meaning that low $R_i^v$ is caused by some property at the scene level. As with urban scenes, slope is also a problem, usually below 0.95, and reaching as low as $S_{18}^v = 0.760$. (For band 13, slope is worse, $S_{13}^v = 0.385$, but this is explained by the spectral response function and the O2 A-Band).

To explain the poorer comparisons of RTOA for urban and vegetation scenes, the effect of the bidirectional reflectance distribution function (BRDF) is considered. A strong back-scattering "hotspot" effect is expected when the viewing zenith and azimuth angles approach the solar zenith and azimuth angles respectively. Sunlight scattered from the surface back to the





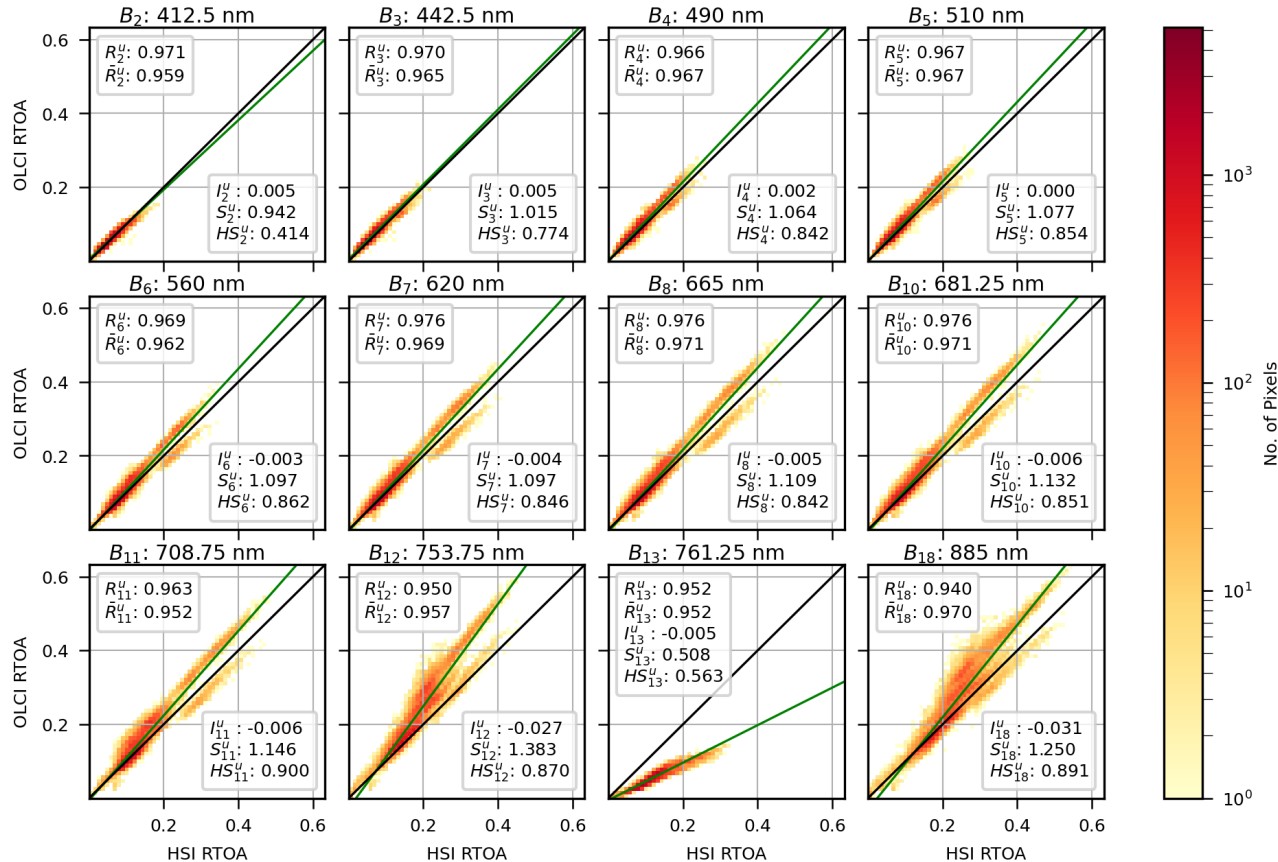

**Figure 4. Urban scene RTOA comparison.** RTOA comparison for 45 urban scenes.

sun does not encounter any surfaces to impede its path. In the general case, as the viewing geometry approaches the solar geometry less surfaces are expected to be encountered that can block lights path to the satellite (Gatebe and King, 2016). Increased complex vertical structures within a scene, such as canopy, is presumed to increase the hotspot effect, as it increases the potential for light scattered in the direction of the observer to encounter another surface. As 20 of the 36 vegetation scenes

are dense vegetation, a strong mean hotspot effect across these vegetation scenes is expected. Further, urban scenes offer less complex structure than trees or crops, but do have large vertical structures (buildings, bridges, etc) increasing the potential for the existance of surfaces blocking light scattered towards the satellite as the viewing geometry deviates from the solar geometry. Though vertical structure is possible for desert scenes, it is a less mandatory feature of such scenes than it is for vegetation or urban scenes, and so the effect of BRDF hotspot over multiple desert scenes on the RTOA reflectance comparison for the two

satellites is expected to be weaker.

To analyse this further a measure of error for the RTOA comparisons is required. For each surface type and each scene, an analysis was performed of the relative distances of both OLCI and HSI viewing geometries from the optimal hotspot geometry. For each scene and band, each instrument is assigned a distance to the hotspot between 0 (if the viewing geometry is exactly





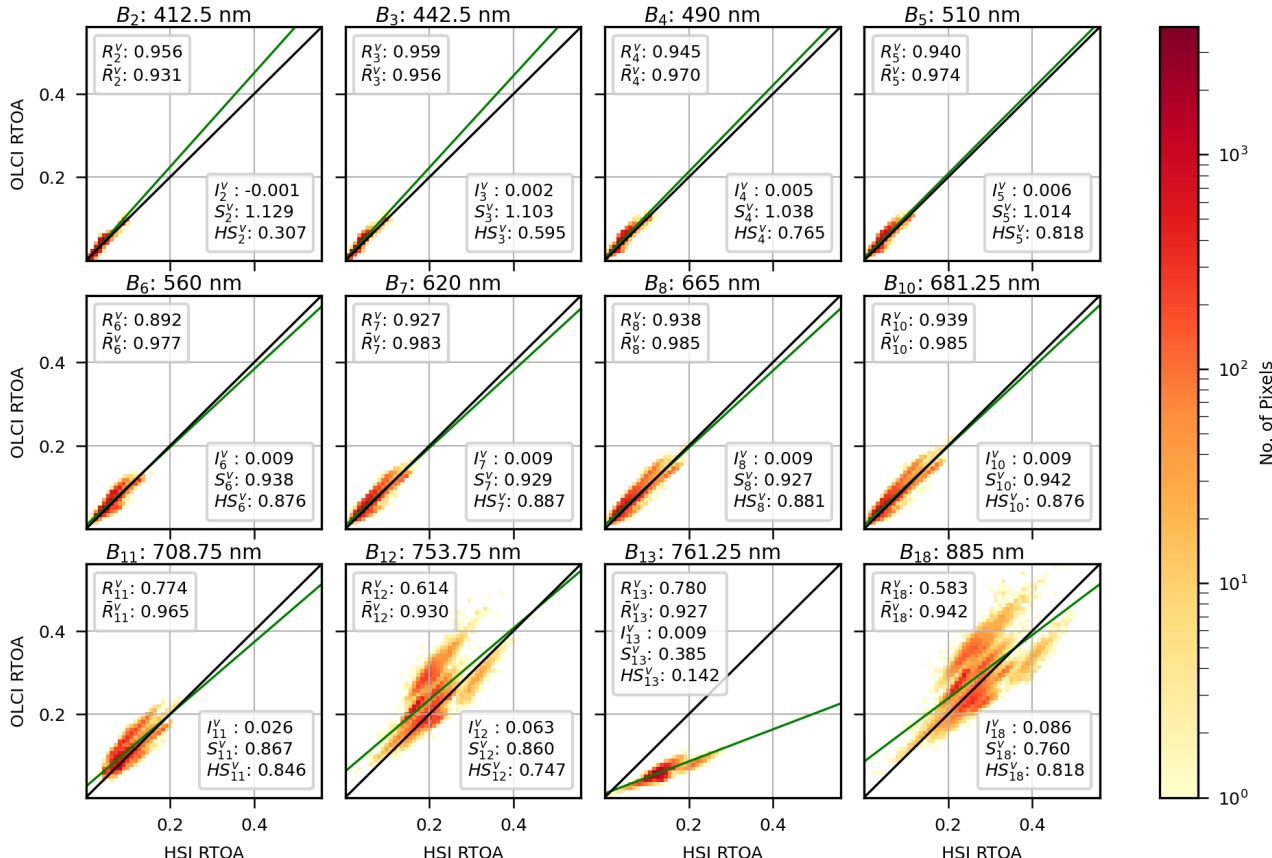

**Figure 5. Vegetation scene RTOA comparison.** RTOA comparison for 36 vegetation scenes.

on top of the hotspot) and 1. A distance of 1 means that both the viewing angles differ from the solar angles by some cutoff amount. The cutoff zenith and azimuth angles are determined per scene type and band as those which maximise the pearson correlation coefficient between absolute difference in the two satellite's distance to the hotspot (a number between 0 and 1) and the root mean squared error (RMSE) per scene. This correlation is presented in Fig. 2, Fig. 4 and Fig. 5 as $HS_i$

With some exceptions, in general $HS_i^d$ is between 0.62 and 0.75, $HS_i^u$ is between 0.84 and 0.9 and $HS_i^v$ is between 0.81 and 0.88. Exceptions to this include $B_{13}$ for which O2 A-band is already a known major source of error. $B_{12}$ is below or at the low end of the range for desert and vegetation scenes where the O2 A-band contributes to the error. Interestingly $HS_{12}^u$ is not affected by this. The lower bands are also exceptions for all scene types with $HS$ decreasing steeply from $HS_4$ to $HS_3$ to $HS_2$. The analysis was repeated for these bands without applying Rayleigh correction to the RTOAs. This resulted in $HS$ for the lower bands similar to that of the higher bands, for each scene type.

In general we see that RMSE most correlates with a hotspot effect for urban scenes (0.84 and 0.9), followed only slightly behind by vegetation scenes (0.81 and 0.88). Desert scenes have the least correlation, but still show some correlation (0.62 and 0.75). Note that we cannot draw any conclusion about a hotspot effect for any scene from this data. We can only conclude that



some mean hotspot effect emerges across scenes of similar type, and this correlates with error in the RTOA comparison. This correlation is strong for urban scenes and vegetation scenes, and weaker for desert scenes. This gives us no information about how strong a BRDF hotspot effect may be for any one scene. The low $HS_i^d$ only demonstrates a weak contribution of any mean hotspot effect that emerges from multiple desert scenes to overall error, but says nothing about how strong hotspots may be for desert scenes. But desert scenes have the best comparison of RTOA (with $R_i^d$, $S_i^d$ and $I_i^d$ deviating least from the ideal), and desert scenes are where a mean hotspot effect of smallest magnitude is expected. Urban scene RTOA comparisons perform worse than desert, and we see that $HS_i^u > HS_i^d$. A stronger mean hotspot effect is expected to emerge for urban scenes than for desert scenes, as urban scenes are expected to have more vertical surfaces which can impede the path of scattered light to a satellite not directly between the sun and the surface. The effect of canopy and crops on Vegetation scenes are also expected to produce a strong hotspot effect, and vegetation scenes do have more error than urban scenes, and similar $HS$.

Finally on BRDF, for all three scene types HSI tends to underestimate OLCI (the exception being $B_{13}$ where spectral response functions ensure HSI overestimates OLCI). For all scene types, the mean distance of OLCI to a hotspot is closer than the mean distance of HSI, which would contribute to OLCI producing higher radiances. Thus two sources of error between HSI and OLCI RTOA are identified; the effect of the O2 A-Band on $B_{13}$ (and to a lesser extent $B_{12}$), and the effect of BRDF hotspots on all bands.

## 4 Application of XBAER to HSI at 10 km Resolution

After gaining confidence in the HSI radiometric calibration, and understanding reasons for RTOA differences, XBAER can now be run for HSI and OLCI, and the retrieved SRF and AOD can be compared. As discussed above, the XBAER cloud mask (Mei et al., 2017b) compares RTOA derived from various bands against calibrated threshold values. Updating the cloud mask for use with HSI data is intended as future work, but the very large RTOA differences for $B_{13}$ mean that XBAER identifies nearly every HSI pixel as either cloud or cloud adjacent. For now, and for the rest of this paper, cloud mask checks involving the O2 A-Band are deactivated and only cloud free scenes are considered. Using the same cloud-free scenes and colocated pixels from Section 3, Fig. 6 presents OLCI vs HSI scatter plots for XBAER-derived SRF and AOD at 550 nm. The spatial resolution of the XBAER output is 10 km. This means that each 30 km scene will have a maximum of 9 output pixels. In the previous section the mean scene correlation ($\bar{R}_i^j$) for $B_i$ was used as a central measure of scene correlation ($R_i^j$) for scene j and $B_i$. The small number of output pixels per scene results results in more outliers for scene correlation of SRF ($R_{srf}$) and AOD ($R_{aod}$). Thus the median will now be used as a central measure of scene correlation ($med(R_{srf})$, $med(R_{aod})$) rather than the mean.

The SRF comparison is very good, with $R_{srf} = 0.953$, $I_{srf} = 0.003$ and only the slope deviating slightly from what would be ideal; $S_{srf} = 1.082$. For higher surface reflectances (predominantly desert scenes, but some urban scenes too) HSI underestimates OLCI and this is generating the slope. The AOD comparison has $R_{aod} = 0.809$, $I_{aod} = 0.153$ and $S_{aod} = 0.785$. Most AOD values are low, in the region of the graph before the red best fit line intersects the black 1:1 line, thus HSI underestimates OLCI for AOD too.





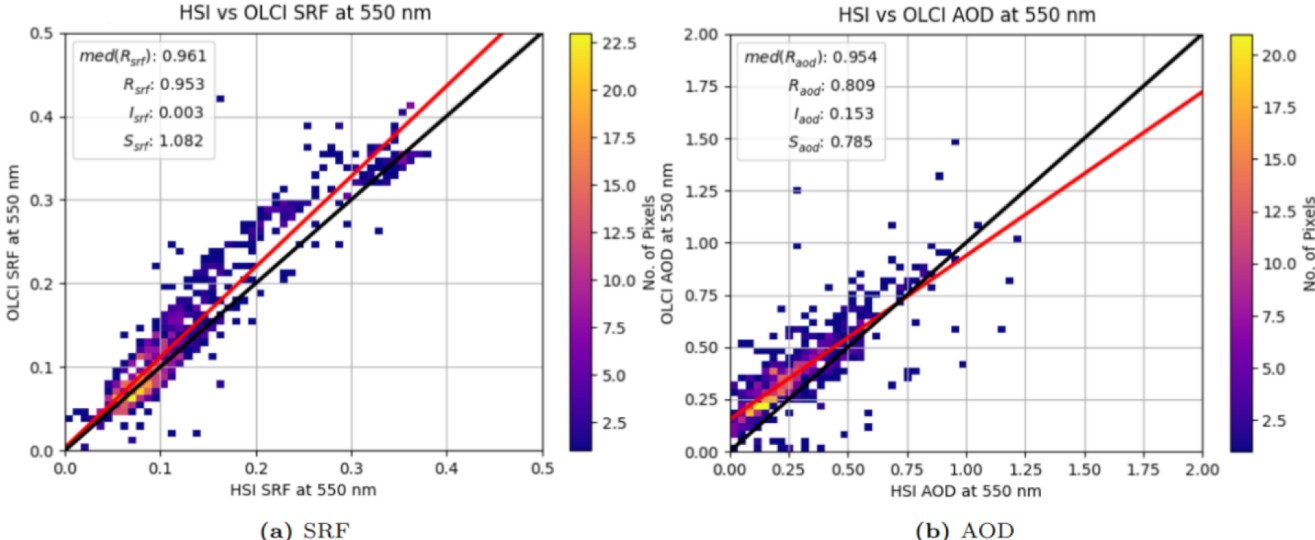

**Figure 6. OLCI vs HSI SRF and AOD.** Comparison of XBAER-derived SRF and AOD at 550 nm using all colocated, cloud-free HSI and OLCI scenes. Retrieved SRF and AOD are at 10 km resolution.

Separating these graphs by scene type may allow us to generate insights in this result. Figure 7 separates the graphs of Fig. 6 by scene type. For SRF the desert scenes perform best with $R^d_{srf} = 0.951$, $I^d_{srf} = 0.024$ and a slope slightly below 1; $S^d_{srf} = 0.997$. This appears to contradict the SRF comparison for all scenes where HSI's underestimation of OLCI appeared to be stronger for brighter scenes ($I_{srf} = 0.003$, $S_{srf} = 1.082$). It turns out it is the brighter of the urban scenes (cities within dry, bright landscapes) which are contributing to the high slope of SRF, with $I^u_{srf} = -0.018$, $S^u_{srf} = 1.245$. In spite of this, SRF

correlates well for urban scenes, $R^u_{srf} = 0.896$. The darker vegetation scenes also contribute to the high slope for all scenes, although less strongly than urban scenes with $I^v_{srf} = 0.003$, $S^v_{srf} = 1.073$. Vegetation scenes also have the lowest correlation for SRF; $R^v_{srf} = 0.75$.

   This SRF comparison for the different surface types is explained using the $B_{18}$ RTOA comparisons from the previous section. $B_{18}$ plays a very important role in retrieving SRF in XBAER. XBAER treats the surface contribution of the signal as a linear

function with SAVI as the variable (Mei et al., 2017a, 2018). $B_{18}$ is used to calculate SAVI, thus a poor RTOA comparison of $B_{18}$ between HSI and OLCI will have an effect on the SRF comparison between HSI and OLCI at any wavelength. Hence the effect of BRDF hotspots on the comparison of $B_{18}$ can propagate to the SRF comparison at any wavelength. It can be seen that the desert, urban and vegetation correlations for $B_{18}$ and SRF follow a similar trajectory; 0.978, 0.940 and 0.583 respectively for $B_{18}$, and 0.951, 0.896 and 0.75 respectively for SRF. It can also be seen that the median scene correlation for desert, urban

and vegetation scenes are all high; $med(R^d_{srf}) = 0.957$, $med(R^u_{srf}) = 0.943$ and $med(R^v_{srf}) = 0.973$, meaning that pixels within a scene tend to be very well correlated, and that what error does exist is primarily introduced by varying properties at the scene level, such as relative distance of the two satellite instruments to a BRDF hotspot.





**Figure 7. OLCI vs HSI SRF and AOD by scene type.** Comparison of XBAER-derived SRF and AOD at 550 nm using colocated, cloud-free HSI and OLCI vegetation, desert and urban scenes. Retrieved SRF and AOD are at 10 km resolution.





Another explanation for the SRF comparisons of different scene types is the difficulty in separating atmospheric and surface contributions from satellite data. The theory behind the XBAER algorithm uses the Chandrasekhar equation for RTOA (Mei et al., 2017a; Kaufman et al., 1997) which separates RTOA into surface and atmospheric contributions. The difficulty in separating the surface and atmospheric signals produces error. The larger the signal to be removed, the larger this error will be. AOD retrieval over bright surfaces (desert, snow) introduces more error than AOD retrieval over dark surfaces (water, vegetation). The inverse is also true; retrieving the surface contribution introduces more error for darker scenes where the atmospheric signal is a higher proportion of the total. Further, this error should be similar for all pixels of the same scene (assuming similar viewing geometry and atmospheric effects for all pixels, which is reasonable for small HSI scenes), but different for pixels of different scenes of similar surface type (assuming dissimilar viewing geometry or dissimilar atmospheric effects), reducing the correlation for all pixels in all scenes, but not reducing correlation for pixels of a single scene. And the darker vegetation scenes have the lowest overall correlation of SRF ($R^v_{srf} = 0.75$) and the brighter desert scenes have the highest overall correlation ($R^d_{srf} = 0.951$).

Similarly, the brighter desert scenes should have the highest error in AOD retrieval, and this is the case with $R^d_{aod} = 0.645$, $I^d_{aod} = 0.168$, $S^d_{aod} = 0.711$. There are also more desert scenes (50) than any other scene type (45 urban scenes and 36 vegetation scenes), so this has a downward effect on the overall comparison of AOD for all scene types; $R_{aod} = 0.809$, $I_{aod} = 0.153$ and $S_{aod} = 0.785$. Nevertheless, the AOD for desert scenes are not enough to explain the performance of AOD for all scene types. As discussed, vegetation scenes ($R^v_{aod} = 0.747$, $I^v_{aod} = 0.161$, $S^v_{aod} = 0.834$) are expected to have the largest BRDF hotspot effect, and this is expected to propagate into the AOD retrieval. Urban scenes perform better ($R^u_{aod} = 0.930$, $I^u_{aod} = 0.132$, $S^u_{aod} = 0.836$) but HSI still underestimates OLCI, and as with vegetation scenes a BRDF hotspot effect may be responsible. Nevertheless, the overall results (Fig. 6) are encouraging.

## 5 Application of XBAER to HSI at 30 m Resolution

Having demonstrated good results at 10 km resolution, as a trial, XBAER is used to retrieve AOD at the full 30 m spatial resolution of HSI. At present, with the exception of the removal of cloud mask checks related to cloud top height, no changes to XBAER have been made to reflect the higher spatial resolution nature of the input data. The purpose of this exercise is to demonstrate the potential for high spatial resolution AOD retrieval using XBAER.

As a first qualitative exercise, a single scene on the Myanmar/Thailand border with visible smoke plumes is selected for closer examination, see Fig. 8 and Fig. 9. The scene contains Thailand's Doi Pha Hom Pok national park, a more forested upland region running north to south through the center of the scene, and also extending to the north east. The west (Myanmar) and south east (Thailand) of the scene is lower land with higher levels of human activity and these are the regions of the scene for which AOD has been retrieved.

An exception to this is two biomass burning events (Fig. 9) in the center of the scene, with expected aerosol type of black and brown carbon. The more westerly fire's source is across the Thai border in Myanmar, and so not in the national park, but





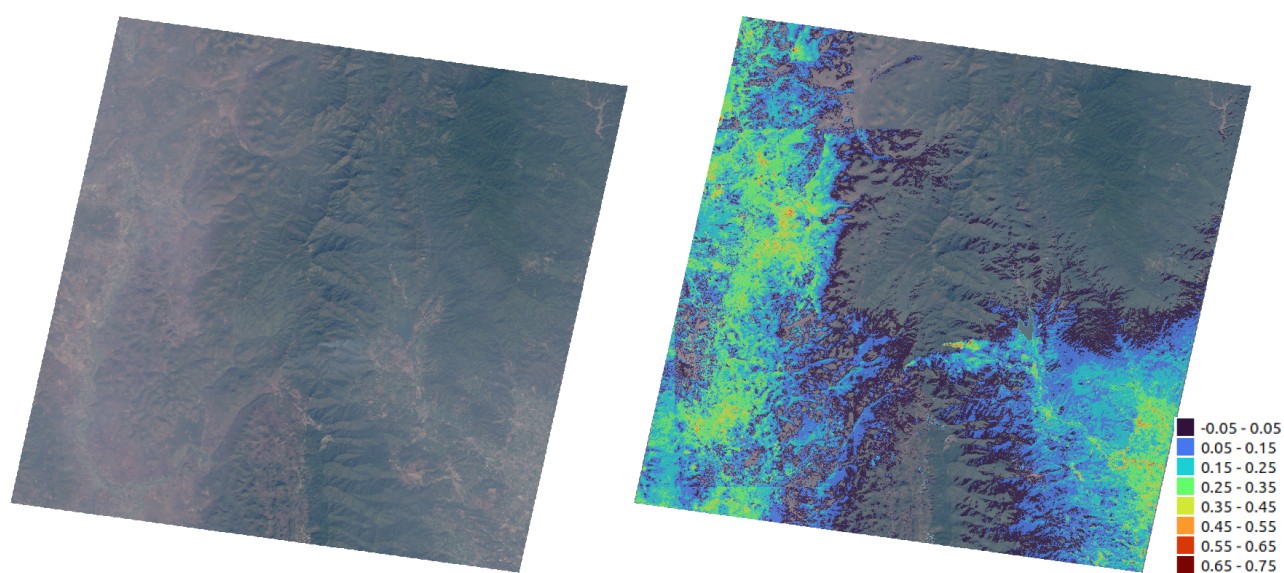

**Figure 8. High resolution XBAER AOD.** Scene on the Myanmar/Thailand border. (a) HSI RGB Image of the VNIR scene provided as part of the EnMAP data. EnMAP data ©DLR [2023] All rights reserved. (b) 30 m spatial resolution, XBAER-retrieved AOD.

still in a relatively forested and upland region that tends to be characterised by low AOD. In general the area of the national park has no or very low AOD retrieved, but the plumes of the fires are the most obvious exception to this.

A more quantitative exercise is a comparison of the 30 m XBAER AOD with AERONET Level 2.0 AOD at 550 nm, Fig. 10. There are 116 colocations from 8th June 2022 to 15th April 2024 for a variety of surface and aerosol types. Retrievals of negative AOD are excluded, as well as AOD above 2. The overall comparison has $R = 0.631$, $I = 0.085$ and $S = 0.724$. 85%

of the colocations have low aerosol loading ($AOD < 0.3$). Presumably high AERONET/low XBAER colocations are due to XBAER not being adapted for high resolution HSI data, and high XBAER/low AERONET AOD are due to cloud contamination. Some cloud contamination is expected as the XBAER cloud mask has had the cloud top height checks deactivated (as discussed in the previous two sections), and other threshold values have not yet been calibrated to HSI radiances.

An attempt has been made to perform the colocations in line with the spatial-temporal technique of Ichoku et al. (Ichoku

et al., 2002), but the small HSI scene size places restrictions on this. Temporally, AERONET observations are averaged for the hour period centered at the time of the HSI measurement. Spatially, all HSI pixels within 10 km of the AERONET site and within one standard deviation of the mean for that radius are averaged. Typically such colocations are performed over a much wider spatial area, but a HSI scene is 30 km and AERONET sites are more frequently closer to the edge of the scene than the center. Usually the full 10 km radius circle around the AERONET site is not fully within the scene. Further, to increase the

amount of colocations, the AERONET site is allowed to be as much as 1 km outside the scene.





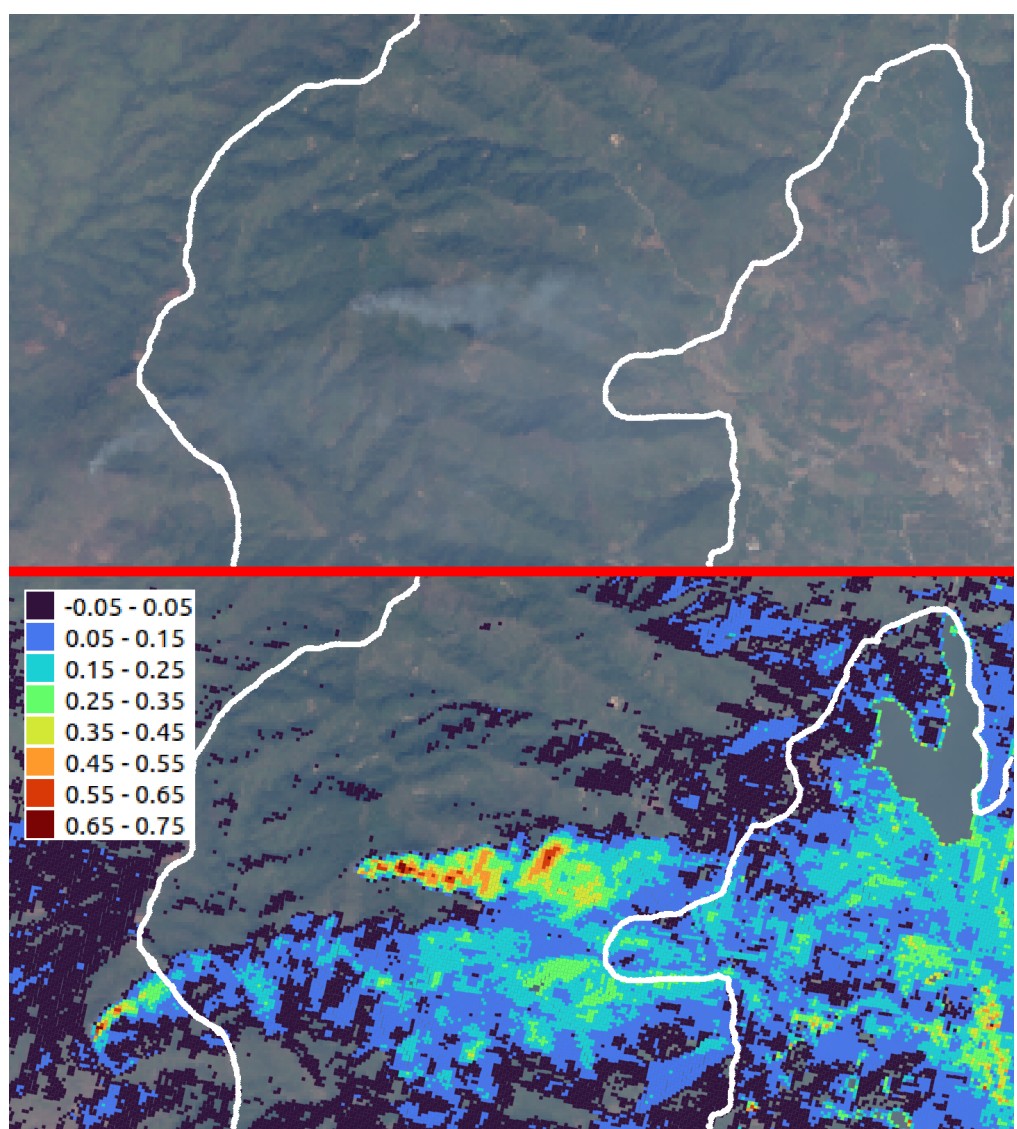

**Figure 9. Biomass burning plumes.** Two biomass burning events are visible in the RGB image (top, EnMAP data ©DLR [2023] All rights reserved); one in the center, one in the south west. The plumes are visible in the AOD retrieval (bottom). The white lines indicate the borders of the Doi Pha Hom Pok national park. The western border is also the Myanmar/Thailand border.



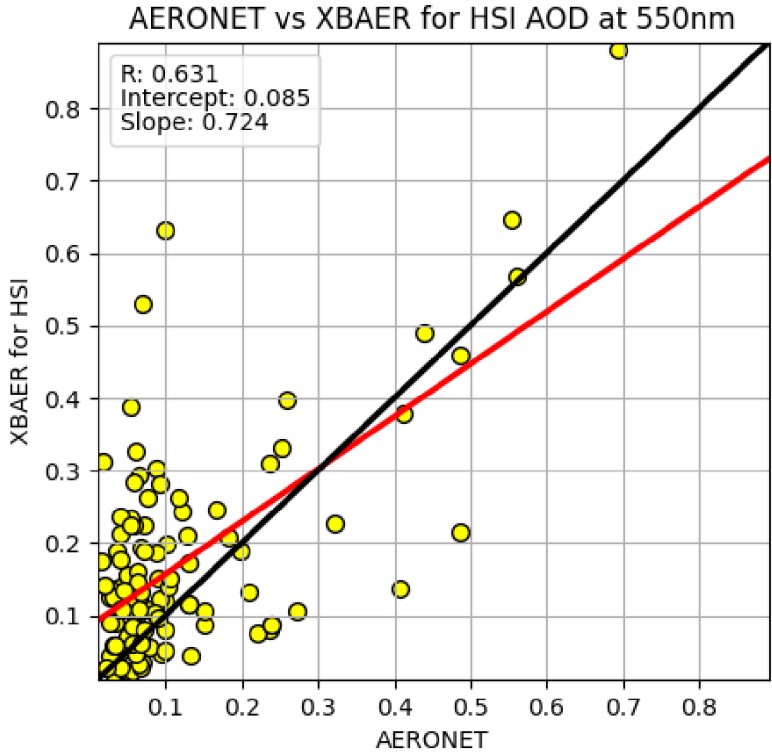

**Figure 10. XBAER AOD vs AERONET.** Preliminary 30 m spatial resolution XBAER AOD retrieval vs AERONET at 550 nm.

## 6 Conclusions

In general, a very good comparison of RTOA between colocated OLCI and HSI pixels was demonstrated for a variety of surface types. Bands and surface types where good comparisons (Pearson correlation coefficient > 0.95, intercept and slope of the best fit line within 0.05 of 0 and 1 respectively) were not found could be ascribed to one of two reasons; 1) differences in
spectral response functions at or near the O2 A-Band, and 2) relative difference of the two satellite viewing geometries from the expected location of a BRDF hotspot. The spectral response function over the O2 A-Band explain the bad comparison between OLCI band 13 and HSI band 62. This reason also contributes to a less than optimal fit between OLCI band 12 and HSI band 61. BRDF hotspots can occur when the viewing geometry is the same as the solar geometry; increased reflectance is measured by the viewer as this is the only direction in which all sunlight scattered from the surface to the viewer encounters
no other surface blocking its path. A strong mean hotspot is expected across all urban scenes and all vegetation scenes, and in general there is a high Pearson correlation coefficient between the RMSE of the RTOA comparison and the relative distance of the two satellites from the hotspot location; generally between 0.84 and 0.9 for the urban scenes, and between 0.81 and 0.88 for the vegetation scenes. Desert scenes are expected to have a lesser mean hotspot effect than urban and vegetation scenes,



and as well as having the best RTOA comparison, they also have the least correlation between RMSE and the relative distance
of the two satellites from the hotspot location; generally between 0.62 and 0.75.

These colocated OLCI and HSI pixels were then used to retrieve SRF and AOD using XBAER at 10km spatial resolution.
SRF in particular had a very good comparison with correlation of 0.953 and a best fit line with intercept of 0.003 and slope of
1.082. Separating the comparison by scene type it was shown that desert scenes performed best, but vegetation and urban scenes
hurt the comparison. BRDF hotspot effects which most affect vegetation and urban scenes are believed to be the cause. The
AOD comparison is not as strong as the SRF comparison, with correlation of 0.809 and a best fit line with intercept of 0.153 and
slope of 0.785. These results are particularly impacted by reduced performance over the more numerous bright desert scenes.
The consequences of BRDF hotspots on SRF retrieval may also be hurting AOD retrieval for urban and vegetation scenes.

The 30 m spatial resolution XBAER AOD was able to identify plumes from biomass burning in a local region that otherwise
would be expected to have low AOD. A comparison of multiple scenes with AERONET AOD produced good results, consid-
ering that XBAER has not yet been modified to produce data at this resolution; ($R = 0.631$, $I = 0.085$ and $S = 0.724$). This
points the way to necessary modifications to XBAER that will allow it to produce high spatial resolution data products with
higher accuracy.

The cloud top height threshold of the cloud mask needs to be modified to handle the much wider HSI spectral response
functions near the O2 A-Band. The other cloud mask threshold values (for scene brightness and homogenity) do not require
as big a change, but still need to be modified. In particular the effects of viewing geometry on BRDF hotspots may need to be
taken into account in XBAER's surface treatment. Lastly an analysis of how best to compare surface-based measurements to
satellite retrievals is needed as the very small HSI scene size makes the spatial-temporal technique of Ichoku et al. ((Ichoku
et al., 2002)) difficult.

*Author contributions.* LM and MV conceived the research. SL processed the data and wrote the manuscript. LM, MV and HB helped in
shaping this paper. LM and MV acquired the funding. All the authors contributed to the interpretation of the results and the final drafting of
the manuscript.

*Competing interests.* At least one of the (co-)authors is a member of the editorial board of Atmospheric Measurement Techniques.

*Acknowledgements.* The authors would like to express their gratitude to the German Aerospace Center (Deutsches Zentrum für Luft- und
Raumfahrt e.V., DLR) and the German Federal Ministry for Economic Affairs and Climate Action (Bundesministerium für Wirtschaft und
Klimaschutz, BMWK) who have supported this research. Thanks also to Martin Bachmann and DLR as well as Maximilian Brell and GFZ
Potsdam for help interpreting the EnMAP data.



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
