# Peer review of "Application of XBAER aerosol optical depth retrieval algorithm to hyperspectral EnMAP satellite data"

_EGUsphere, 2025_

## Author Response (AR2)

**POINT BY POINT RESPONSE TO REVIEWER 1**

**Dear Reviewer, here is a more complete response to your comments, in which we also outline what actions have been taken in the latest draft which will be uploaded today. We print your entire comment in plain text, with our responses in bold where relevant**

In this study, the authors apply the eXtensible Bremen AErosol Retrieval (XBAER) AOD retrieval algorithm to data from the Environmental Mapping and Analysis Program (EnMAP) satellite. The retrieved aerosol optical depth (AOD) values are compared with observations from the Ocean and Land Colour Instrument (OLCI). In addition, biomass-burning plumes are identified within a single scene, and a comparison with AErosol RObotic NETwork (AERONET) AOD measurements is conducted across multiple scenes. The results demonstrate strong agreement among these datasets.

The study also highlights that discrepancies between measurements can occur due to differences in the spectral response functions of the instruments, particularly in the region around the $O_2$ A-band absorption feature or due to varying viewing geometries, which influence the bidirectional reflectance distribution function (BRDF) effects.

It is worth noting that working with EnMAP satellite data poses significant challenges, particularly due to the lack of geo-referencing, which likely required considerable effort and time during data processing for the authors.

This manuscript presents valuable findings at meter-scale resolution by leveraging the XBAER algorithm and EnMAP data. The work provides important scientific insights that contribute to the global monitoring of high-resolution AOD and biomass-burning plume detection.

The manuscript is well-written, clearly structured, and aligns with the aims and scope of the Atmospheric Measurement Techniques (AMT) journal. I recommend its publication, subject to minor revisions and a few general suggestions.

I will start with general suggestions:

1. The title is "First results of the XBAER…..", my question if this is the first result then what is/will be the second result, third result so on….?

In general the title of the manuscript should reflect the content as well as should be catchy for the scientific audience to read it and so some applications as well. So, my suggestion would be something like (Application of XBAER aerosol optical depth retrieval algorithm to EnMAP satellite observations).

I feel like this title would reflect the content as well as it will develop the interest among the scientific community to read it. However the are authors free to think about this.

**In the new draft we have changed the title to "Application of XBAER aerosol optical depth retrieval algorithm to hyperspectral EnMAP satellite data" Another reviewer commented that EnMAP wasn't well known so we also include the term "hyperspectral" to be more specific about what EnMAP is.**

Minor comments:

1. Line 16: "Future modifications to XBAER that would allow it to produce more accurate retrievals at HSI's spatial resolution are discussed". This sentence is not needed in the abstract, remove it.

**This has been removed**

2. In introduction please cite some literatures, in line 19- 21.

**Lines 19-21 are now Lines 18-20 and run as follows: "Aerosol optical depth (AOD) is a measure of atmospheric aerosol, defined as the columnar integration (between surface and top of atmosphere) of the extinction coefficient (the sum of the absorption and scattering coefficients) of aerosol (Di Antonio et al., 2023)"**

3. Line 51: "This paper investigates the possibility of adapting the XBAER algorithm for use with HSI data", Mention that the combinations of XBAER and EnMAP data would be valuable to detect fine resolution small scale air pollution and bio-mass burning plumes over different regions.

**As Lines 36-37 discussed motivations for high resolution AOD, we have changed those lines (now at lines 38-39) to mention this. The new line is as follows: "While these AOD datasets are invaluable, higher resolution AOD could reveal regional small scale air pollution and bio-mass burning plumes, which can help improve exposure studies of the health effects of aerosols."**

4. At methods and Data section, at line 85-89, please mention advantage of using EnMAP satellite observations over all other sensors mentioned above, and its further alignment with the aim/objective of the manuscript.

**Lines 85-86 previously read "HSI on board the EnMAP satellite is a pushbroom hyperspectral imager and measures radiance data at 30 m, ten times the full spatial resolution of OLCI. ". We have changed this to Lines 91-93: "Like MERIS and OLCI, HSI on board the EnMAP satellite is a pushbroom imager. Unlike MERIS and OLCI it is a hyperspectral imager with 224 bands. More importantly for the purposes of high-spatial resolution AOD retrieval, HSI measures radiance data with a spatial resolution of 30 m, ten times the full spatial resolution of OLCI and MERIS."**

5. The results sections are very well written and easy to follow.

6. In figure 1, these stations are used for global validation, so is it possible to use present the AOD retrieval from the combination of XBAER and EnMAP for the global region? If yes, then present the figure and discuss it. If no, then mention it why?

**Lines 93-95 are new and explain that the drawback of high spatial and spectral resolution is low global coverage: "The drawback to high spatial and spectral resolution is low daily global coverage (with a swath width of 30 km and a daily limit on swath length of approximately 5000 km), making global retrievals of dynamic variables with HSI data not possible."**

7. As Arctic is warming much faster than global average and the new method of retrieval of AOD is very much needed. This would be very nice to create a separate section after conclusion, may be discussion section, and mention how this entire framework of retrieval would be helpful for different regions undergoing rapid climate change particularly Arctic region, and the future application aspects of this study.

**Unfortunately in this study we wished to avoid retrievals of snow and ice and regions where those surface types are likely. As you correctly say what is happening in the Arctic is very important, but we should not make any link between our study and that region.**

8. Is it possible to improve the plot quality, particularly the spatial plots?

**We have updated Figure 8 and Figure 9 with a heading for the Legend, also mentioned which wavelength the AOD was retrieved for. For Figure 8 we have also included latitude and longitude for the left hand image.**

I hope these suggestions would be helpful for the further improvement of this study and would be easier for the potential readers to follow. At the end, I would like to mention that this manuscript very well written and well structured and I enjoyed reading it.

**Thank you for your kind words and very useful suggestions. Please let us know if there is anything we failed to address.**

**POINT BY POINT RESPONSE TO REVIEWER 2**

**Dear RC2**

We thank you for your reply and the time you spent on it, as well as your kind words about the manuscript. Following your comments and the comments of your fellow reviewers, a new draft of the manuscript will be uploaded later today

Below is our reply. We include the entire text of your comment in plain text, and use bold text to reply to each point:

In this paper presents an extensive analysis of the eXtensible Bremen AErosol Retrieval (XBAER) algorithm for aerosol optical depths (AOD) retrievals at very low spatial resolution using passive satellite data. The paper is based on look-up tables with is a well-known technique in the retrieval of AODs from space radiometry. Authors present a very interesting comparison of reflected radiance at the top of the atmosphere for OLCI and the hyper-spectral imager on board the Environmental Mapping and Analysis Program. Results of this comparison are very well discussed and presented, being easy to follow the main points remarked by the authors. The comparison is extended to retrieved AODs product with XBAER algorithm, and authors have a great discussion about the agreements between both products and the causes of disagreement when it happens. The XBAER algorithm is later applied to hyper-spectral imager with 30 m resolution, and coherent retrievals are observed for the study cases presented.

Overall, the paper looks very good and the topic is very interesting, and therefore I recommend the publication in AMT.

Finally, the 30 m resolution products are validated versus reference ground-based AERONET measurements. This might be the weakest part of the manuscript as it was not clear for me to see the difficulties remarked for evaluating the 30 m resolution retrievals. I highly recommend re-making this part making clearer how the matchups are made.

**An extensive rewrite of the two paragraphs from Lines 267-280 has been performed. Previously the first paragraph introduced results and then the second paragraph discussed how matchups were performed. The two paragraphs are now at Lines 278-299. The first paragraph now describes how the matchups are performed in much more detail, and the second paragraph describes the results. These two paragraphs now run as follows:**

**START**

**A more quantitative exercise is a comparison of the 30 m XBAER AOD with AERONET Level 2.0 AOD at 550 nm, Fig. 10. The spatial-temporal colocation scheme of Ichoku et al. (2002) is often used to compare spatially-averaged satellite measurements with temporally-averaged ground site measurements. We follow their temporal averaging scheme for a ground site by averaging AERONET measurements for the 1 hour period centered on the HSI overpass time. Their spatial averaging scheme presents us with difficulties. The smallest spatially-averaged region investigated by Ichoku et al. was 30 km × 30 km which they rejected as their pixel size was 10 km × 10 km, and the nine pixels of a 30 km × 30 km region was considered too small a sample. As the HSI pixel size is 30 m × 30 m, this does not present us with a problem. However, as HSI scenes are of area 30 km × 30 km, and as only a minority of colocated**

AERONET sites are near the center of a scene, we reject 30 km × 30 km as a spatially averaged region. Instead we choose a circular region of radius 10 km centered on the AERONET site. Further, as it can be expected that 8 out of every 9 AERONET sites within a HSI scene will be within 10 km of the scene border, we do not exclude colocations where some of the 10 km radius circle is outside the HSI scene. Due to the sparse daily global coverage of HSI, colocations with AERONET sites are rare, and so to increase the amount of colocations further, we include matchups where the AERONET site is as much as 1 km outside the HSI scene. In this way we found 116 cloud-free colocations from 8th June 2022 to 15th April 2024 for a variety of surface and aerosol types. Retrievals of negative AOD (which are possibly due to uncertainties in surface parameterization and lookup table resolution) are excluded, as well as AOD above 2. To further exclude outliers, spatial averaging is then only performed across pixels which are within one standard deviation of the mean.**

**The overall comparison has R = 0.631, I = 0.085 and S = 0.724. 85% of the colocations have low aerosol loading (AOD < 0.3). Presumably high AERONET/low XBAER colocations are due to XBAER not being adapted for high resolution HSI data, and high XBAER/low AERONET AOD are due to cloud contamination. Some cloud contamination is expected as the XBAER cloud mask has had the cloud top height checks deactivated (as discussed in the previous two sections), and other threshold values have not yet been calibrated to HSI radiances.**

**END**

Another point I want to remark on is that the title is confusing as not everybody is familiar with the term 'EnMAP', and thus I highly recommend finding a more suitable title. Apart of that, I have some minor comments that might help authors to improve their manuscript.

**We have made an extensive change to the title, however we did wish to keep EnMAP in the title. To make it clear to readers what EnMAP is, we now describe EnMAP as a hyperspectral satellite within the title. The full title now runs as "Application of XBAER aerosol optical depth retrieval algorithm to hyperspectral EnMAP satellite data"**

Introduction: My general comment is that authors do not discuss the advantages of an algorithm based on look-up tables versus that retrieve aerosol and surface properties using algorithms based on least-squares minimization methods. That should be acknowledged in the introduction.

**Lines 74-76 have been added in the new draft, stating "The use of lookup tables of appropriate resolution increases the speed of the algorithm over one which must do its own radiative transfer modeling, while preserving sufficient accuracy."**

Lines 19-21: I think that a more clear and concise definition of AOD is needed. References are also required.

**The Lines have been updated. Previously Lines 19-21 stated "Aerosol optical depth AOD is a measure of how much atmospheric aerosols weaken the transmission of radiation through the atmosphere, and thus acts as a proxy measure for the amount of aerosol in the atmosphere. Retrieving AOD thus helps us to understand how aerosols affect climate, weather and health."**

**These have now been updated as Lines 18-20 and state "Aerosol optical depth (AOD) is a measure of atmospheric aerosol, defined as the columnar integration (between surface and top**

of atmosphere) of the extinction coefficient (the sum of the absorption and scattering coefficients) of aerosol (Di Antonio et al., 2023). "

Lines 22-25: AERONET is based on sun-photometry measurements from the ground and is usually the reference versus space measurements. This should be highlighted. Also, I recommend adding in the discussions other airborne field campaigns.

**Lines 21-23 have been updated. They previously stated "The most accurate measurements for AOD come from sparse surface-based measurements, such as the various sites of the AErosol RObotic NETwork (AERONET) (Holben et al., 1998)."**

**These have now been updated as Lines 20-22 and state "The most accurate measurements for AOD come from sparse surface-based measurements, such as the numerous sun-photometry measurement sites of the AErosol RObotic NETwork (AERONET) (Holben et al., 1998), which are often used as comparison to space-based retrievals."**

**Regarding airborne field campaigns; Lines 23-25 previously stated "Airborne instruments can provide extended spatial coverage, for example AOD has been retrieved using the Airborne Visible/Infrared Imaging Spectrometer (AVIRIS) data (Isakov et al., 1996), but such measurements can only be performed for infrequent campaigns".**

**These are now updated as Lines 22-25 and state: "Airborne instruments can provide extended spatial coverage, for example AOD has been retrieved from Airborne Visible/Infrared Imaging Spectrometer (AVIRIS) data (Isakov et al., 1996), the AerosolCloud Meteorology Interactions over the Western Atlantic Experiment (ACTIVATE) mission (Sorooshian et al., 2025), and the KORea–US Air Quality) atmospheric experiment (KORUS-AQ) (LeBlanc et al., 2022). However airborne measurements can only be performed during infrequent campaigns."**

Methodology: The filtering of cloud-affected data is something that I could not understand after reading the manuscript. Can the authors give a better explanation?

**The paragraph on cloud masking has been updated to be explicit that a full description of the XBAER cloud mask is given in the paper by Mei et al., 2017b. The new paragraph (Lines 64-68) runs now as: "The XBAER retrieval begins by applying a cloud mask to minimize cloud contamination in AOD retrieval. Clouds are identified by comparing measures of scene-brightness, RTOA homogeneity and cloud height information to calibrated threshold values. The threshold values are determined by a combination of radiative transfer modeling and an analysis of different cloud, aerosol and surface scenarios. Cloud adjacent pixels are also screened. A full description of the cloud mask is given in Mei et al. (2017b)."**

Line 72: What are the different aerosol types and their properties used in the look-tables computations?

**This information is in one of our references. A new line has been added in this paragraph (now at Lines 69-78) making it explicit that this reference contains this information: " A full description of the XBAER algorithm's processing chain, surface treatment, lookup tables and AOD retrieval can be found in (Mei et al., 2017a)."**

Line 99: I think there are typos. I miss the verb in 'pixel fully within'. Also, typo in 1,600 30 m HIS

**We recognise now that the language used may not have been as clear as it could have been. Previously this line read: "For each 1.2 km OLCI pixel fully within a HSI scene, as much as 1,600 30 m HSI pixels may overlap it either fully or partially."**

**We have updated this line (now at Lines 109-110) to be "For each 1.2 km OLCI pixel which is fully within a HSI scene, as much as 1,600 of the 30 m HSI pixels may overlap it either fully or partially."**

Section 3 Comparisons of OLCI and HIS RTOA: If I understand well, one of the main findings is the influence of surfaces blocking to explain the differences. Why not is mentioned in the abstract?

**We do mention BRDF in the abstract, which includes the effect of surface blocking within a pixel.**

Line 225: What is SAVI? It is not defined.

**We have defined the acronym SAVI in Lines 69-72 as part of the XBAER description. Further details on it exist in the paper referenced on XBAER at the end of the paragraph; Mei et al., 2017a**

Line 237 – 239: I think that references are needed because this is known from other studies for other sensors and algorithms.

**We have included now a reference to H.H. Ku's 1966 paper describing the law of propagation of error, which is the mathematical principle we are describing. We also now correctly name this effect as that law. Unfortunately, while I could find references which described this effect for aerosol phase function, aerosol layer thickness and aerosol single-scattering albedo ("Error sources in the retrieval of aerosol information over bright surfaces from satellite measurements in the oxygen A band", Nanda et al., 2018), I could not find such a reference for AOD retrievals (though we do not imply that they do not exist). There are many papers describing how brighter surfaces make retrieval of the surface difficult (for example for the MODIS dark target AOD retrieval algorithm), which in turn makes AOD retrieval difficult for brighter surfaces, however this is a different effect to what we describe, where hypothetical equal error in all surface retrievals ensures greater error in AOD retrieval for brighter surfaces.**

**Previously Lines 236-237 read "The difficulty in separating the surface and atmospheric signals produces error."**

**This has now been extended as Lines 246-248 with an additional sentence: "The difficulty in separating the surface and atmospheric signals produces error. The larger the signal to be removed, the larger this error will be, following from the law of propagation of error (Ku, 1966)."**

Figures 8 and Figure 9: They need improvement. I recommend adding the coordinates to the figure. Also, I guess that color-bars refer to AOD but it is not indicated. Wavelength of AODs retrieval is not indicated.

**We have updated the legends of both images to specify that what is being shown is AOD at 550 nm. We have also included coordinates for the corner pixels of the left hand image of Figure 8.**

Line 269: How do you explain negative values in AODs retrievals with XBAER ?

**Line 269 previously read as " Retrievals of negative AOD are excluded, as well as AOD above 2."**

**This has now been updated as Lines 291-292 and read as "Retrievals of negative AOD (which are possibly due to uncertainties in surface parameterization) are excluded, as well as AOD above 2"**

Lines 282-283: It is not mentioned which variable presents the good retrieval.

**There are 36 RTOA variables (for 12 wavelengths and 3 surface types). We do mention the two explanations for why some RTOA comparisons perform poorly, and in describing these reasons we describe the important properties that poorly performing RTOAs share, e.g. at or near the O2 A-band, or due to vegetation and urban BRDF hotspot effects. Additionally, the naming convention for these variables that we employ in the main body of the text will not be clear to anyone who reads only the Abstract and Conclusion, so we hesitate to use it in the Conclusion.**

Once again we thank you for your kind words and the time you spent on this

Kind Regards

POINT BY POINT RESPONSE TO REVIEWER 3

**Dear Reviewer 3**

We wish to thank you for the time spent on our manuscript and your positive suggestions

Below we give our reply. We reproduce your entire response in standard font. Where a response by us is required we add new lines in bold font. We hope all will be clear

The manuscript presents a timely and relevant comparison between data from the EnMAP HSI and Sentinel-3 OLCI sensors, with a focus on extending the XBAER aerosol retrieval algorithm to leverage HSI's higher spatial resolution. The study is well-motivated, particularly in light of the increasing availability of high-resolution hyperspectral satellite data for atmospheric applications.

The comparison of top-of-atmosphere reflectances shows strong agreement (R > 0.9), indicating good consistency between sensors, though the identified deviations near the $O_2$ A-band and due to BRDF effects are important and well explained. The retrieval comparisons for surface reflectance (R = 0.953) and AOD (R = 0.809) are encouraging, suggesting that HSI data are promising for future AOD retrievals using XBAER. However, the reduced performance in AOD retrievals using unmodified XBAER on full-resolution HSI data (R = 0.631 vs. AERONET) highlights the need for targeted algorithm adaptation.

Minor revisions are recommended. To improve the manuscript's clarity and impact, please address the following critical points:

- A more detailed explanation of the causes and implications of the differences in AOD retrieval performance between surface types would be valuable.

**We believe we have covered everything about AOD retrieval performance between surface types;**

**We discuss how the law of propagation of error favours AOD retrieval for darker surfaces. We have added a sentence at Line 247-248 naming this effect and referencing it: "The larger the signal to be removed, the larger this error will be, following from the law of propagation of error (Ku, 1966)."**

**A secondary effect we discuss is how SRF retrieval using SAVI parameterization favours surfaces with less BRDF effects. As the comparison of RTOAs between OLCI Band 18 (885 nm) and HSI Band (887.7 nm) is particularly affected by BRDF effects for vegetation scenes (and to a lesser extent for urban scenes), and this error in SRF retrieval is expected to have an effect on AOD retrieval. The stronger the BRDF effect, the larger the SRF error and the larger the error that propagates into AOD retrieval.**

**Finally we mention that the larger number of desert scenes (which perform worst on AOD retrieval) effect the overall retrieval for all surface types.**

- The discussion of future algorithm modifications would benefit from a clearer outline of specific technical challenges and proposed solutions.

**We have modified the final paragraph of the conclusion to be clearer in its intent. We have added an additional desired improvement, that is an improvement in the surface parameterization dataset. We also removed reference to cloud homogeneity and surface**

brightness thresholds used in the cloud mask, as these are of less importance. Previously Lines 308-313 read as

"The cloud top height threshold of the cloud mask needs to be modified to handle the much wider HSI spectral response functions near the O2 A-Band. The other cloud mask threshold values (for scene brightness and homogenity) do not require as big a change, but still need to be modified. In particular the effects of viewing geometry on BRDF hotspots may need to be taken into account in XBAER's surface treatment. Lastly an analysis of how best to compare surface-based measurements to satellite retrievals is needed as the very small HSI scene size makes the spatial-temporal technique of Ichoku et al. ((Ichoku et al., 2002)) difficult."

These have been updated now as Lines 326-333:

"Finally, further development of XBAER is required to improve its retrievals for high spatial resolution EnMAP data. The cloud top height threshold of the cloud mask needs to be modified to handle the much wider HSI spectral response functions near the O2 A-Band. The effects of BRDF may need to be taken into account, particularly regarding its effect on XBAER's surface treatment. That surface treatment itself relies on a surface parameterization dataset with 0.1° × 0.1° spatial resolution and monthly time resolution. The spatial resolution in particular of this dataset can be expected to introduce localised error into a 30 m × 30 m resolution data product. Lastly an analysis of how best to compare surface-based measurements to satellite retrievals is needed as the very small HSI scene size makes the usual spatial-temporal averaging scheme of Ichoku et al. (2002) difficult."

- Including more scenes or seasons in the AERONET comparison could help generalize the findings.

Regrettably the EnMAP's low global coverage means that colocations with AERONET sites are rare. We have updated Section 2: "Methods and Data" and Section 5: "Application of XBAER to HSI at 30 m Resolution" describing EnMAP's low coverage and its effect on potential AERONET co-locations

In Lines 93-95 we now state "The drawback to high spatial and spectral resolution is low daily global coverage (with a swath width of 30 km and a daily limit on swath length of approximately 5000 km), making global retrievals of dynamic variables with HSI data not possible"

On the advise of another reviewer we have rewritten the last two paragraphs of Section 5, and that includes beginning one sentence at Line 288-289 with "Due to the sparse daily global coverage of HSI, colocations with AERONET sites are rare, ..."

Overall, this is a valuable contribution to the field of aerosol remote sensing and highlights the potential of EnMAP HSI for atmospheric applications when appropriate retrieval adaptations are made.

Line by line comments:

1. At line 26, cite some articles to support this statement "Several decades of AOD measurements from satellites are now available that offer increased spatial coverage over both surface-based and airborne measurements."

**We have included a reference to Wei et al., 2020 ("Satellite remote sensing of aerosol optical depth: advances, challenges, and perspectives") which includes a list of past review articles**

2. For the method section is it possible to show a graphical flow chart?

**A graphical flow chart exists in the reference Mei et al., 2017a. We have updated the text (Lines 76-77) to make it more explicit that that paper includes a full description of the algorithm. "A full description of the XBAER algorithm's processing chain, surface treatment, lookup tables and AOD retrieval can be found in (Mei et al., 2017a)."**

3. At line 199, please mention here as well why the authors used XBEAR algorithm, so that the reader gets the flow while reading the manuscript.

**Lines 198-199 previously read as "After gaining confidence in the HSI radiometric calibration, and understanding reasons for RTOA differences, XBAER can now be run for HSI and OLCI, and the retrieved SRF and AOD can be compared."**

**These lines have now been updated as Lines 208-211 and read as: "After gaining confidence in the HSI radiometric calibration, and understanding reasons for RTOA differences, we now wish to compare SRF and AOD retrievals from the two satellites. As XBAER is an established algorithm that can retrieve both, we use it to process the same colocated OLCI and HSI scenes from the previous section, and compare the resulting SRF and AOD"**

4. Please shorten the conclusion and introduction part a bit for easy reading of the manuscript for the potential readers.

**While we agree that this would be useful, unfortunately other reviewers have requested additional information in the Introduction and Conclusion, and so we have not been able to reduce the amount of text in them. At present both Introduction and Conclusion are each a little over a page long, and that is using a single column format with a larger than usual font and large spacing between lines, as provided in the Copernicus Latex template. Their word counts are approximately 750 and 680 respectively.**

5. Is it possible to present the spatial global view of the retrieval? If not then please discuss why, as this is bit important to get the overview that the combination of XBEAR and EnMAP is effective to capture the AOD distribution globally regardless of the background pollution scenarios.

**Unfortunately no, as explained above regarding the low global coverage of EnMAP. As also mentioned above, the Methods and Data section has been updated to indicate that that is not possible. Lines 93-95 now state "The drawback to high spatial and spectral resolution is low daily global coverage (with a swath width of 30 km and a daily limit on swath length of approximately 5000 km), making global retrievals of dynamic variables with HSI data not possible". Note that global retrievals of XBAER with MERIS and OLCI data do exist and are in the referenced papers Mei et al. 2017a and Mei et al. 2018.**
* * *
Once again we thank you for your review

Kind Regards

**POINT BY POINT RESPONSE TO REVIEWER 3**

**Editor's Comment:** Well done for addressing all of the points raised by the reviewers. I agree with the reviewers that it is a well written manuscript and interesting to our readership. A minor technical point: O2 A-Band could have the 2 subscripted throughout

**Response:** The manuscript has been updated with the desired change